# Effect of Arbuscular Mycorrhiza Fungus *Diversispora eburnea* Inoculation on *Lolium perenne* and *Amorpha fruticosa* Growth, Cadmium Uptake, and Soil Cadmium Speciation in Cadmium-Contaminated Soil

**DOI:** 10.3390/ijerph20010795

**Published:** 2023-01-01

**Authors:** Jiahua Sun, Qiong Jia, Yi Li, Kanglong Dong, Shuai Xu, Yanan Ren, Ting Zhang, Jiayuan Chen, Nannan Shi, Shenglei Fu

**Affiliations:** 1College of Geography and Environmental Science, Henan University, Kaifeng 475004, China; 2Key Laboratory of Geospatial Technology for the Middle and Lower Yellow River Regions, Henan University, Ministry of Education, Kaifeng 475004, China; 3Dabieshan National Observation and Research Field Station of Forest Ecosystem, Henan University, Kaifeng 475004, China

**Keywords:** arbuscular mycorrhizal fungi, *Lolium perenne*, *Amorpha fruticosa*, cadmium

## Abstract

Cadmium (Cd) pollution has become aggravated during the past decades of industrialization, severely endangering human health through its entry into the food chain. While it is well understood that arbuscular mycorrhizal fungi (AMF) have a strong ability to regulate plant growth and Cd uptake, studies investigating how they affect soil Cd speciation and influence Cd uptake are limited. We designed a pot experiment comprising two AMF-inoculant groups (inoculation with *Diversispora eburnea* or no inoculation), three Cd concentration levels (0, 5, and 15 mg/kg), and two plant species (*Lolium perenne* and *Amorpha fruticosa*) to study the effect of AMF *Diversispora eburnea* on plant growth, Cd uptake, and Cd speciation in the soil. The results revealed that *L. perenne* exhibited higher productivity and greater Cd uptake than *A. fruticosa*, regardless of AMF *D. eburnea* inoculation. However, AMF *D. eburnea* significantly altered soil Cd speciation by increasing the proportion of exchangeable Cd and decreasing residual Cd, resulting in Cd enrichment in the plant root organs and the elimination of Cd from the polluted soils. Our experiments demonstrate that inoculating plants with AMF *D. eburnea* is an effective alternative strategy for remediating Cd-contaminated soil.

## 1. Introduction

With the rapid increase in the rate of industrialization, increasing amounts of waste gas, waste water, and waste residue that frequently contain heavy metals are being discharged from industries [1,2,3]. The heavy metal cadmium (Cd) has attracted considerable attention because of its strong mobility and high toxicity. Reports from multiple countries (including Japan and China) have shown that Cd enters the human body through the food chain, where it can cause renal dysfunction and other diseases [4,5,6]. Physical and chemical methods are commonly used in Cd remediation [7], but the costs involved in the processes and the adverse effects they have on soil biological properties have caused controversy, and their residues can cause secondary pollution [8,9]. Therefore, biotechnological solutions, such as phytoremediation, are favored because they are environmentally friendly and do not require the use of high-cost technology [10].

Arbuscular mycorrhizal fungi (AMF) are an important functional group of soil microorganisms that can form a symbiotic relationship with 80% of vascular plant species in terrestrial ecosystems [11]. Previous studies have shown that AMF play an active role in removing Cd from soil [12,13], as AMF mycelium significantly increase the absorption of mineral ions (such as phosphorus) and water in plants, which increases the plant biomass, and previous studies have shown that AMF play an active role in removing Cd from soil [14,15,16]. The symbiosis between AMF and plants improved the activity of antioxidant enzymes in plants, which reduces the Cd toxicity [17,18,19]. Furthermore, Wang et al. (2020) found that AMF can change the form of soil Cd, and increase the acid-soluble Cd content, which assists in the removal of soil Cd by plants [20]. Other studies have found that AMF may promote the retention of Cd in the roots, which reduces the transfer of Cd to the shoots of plants through the fixation and isolation of Cd [21,22]. AMF colonization can also enhance plant resistance to Cd by altering the subcellular distribution and chemical forms of Cd in plants [23,24].

*Diversispora eburnea* (*D. eburnea*) is an AMF species commonly found in the Cd mine tailings and it has promising potential for use in heavy metal remediation in soil [25,26]. *Lolium perenne* and *Amorpha fruticosa* are fast-growing plants that have a wide distribution, and they can be strongly enriched in Cd [27,28]. Their synergistic effect with AMF *Diversispora* [29,30] is known to be beneficial for soil Cd pollution remediation [31,32]. In addition, AMF can provide an abundant supply of phosphorus to *Amorpha fruticosa* during nitrogen fixation [33,34]. Therefore, it is likely that compared with *Lolium perenne*, its growth would be comparatively more improved with AMF under Cd stress.

The aim of this study was to investigate the effect of *D. eburnean* on the growth, accumulation, and transfer of Cd in *Lolium perenne* and *Amorpha fruticosa*, as well as the associated changes in Cd chemical species in the soil. We hypothesized that AMF *D. eburnea* inoculation would promote Cd uptake and accumulation by promoting plant biomass and changing the form of Cd within the soil. The results obtained here can enhance our understanding of the mechanisms involved in combining *D. eburnea* with plants to remediate Cd-contaminated soil.

## 2. Materials and Methods

### 2.1. Soil Preparation

The original topsoil (0–20 cm depth) was collected from the campus of Henan University, China (34°49′3″ N, 114°18′38″ E). The parent material of the soil is Yellow River alluvium, and the soil type is predominantly yellow fluvio-aquic soil with a sandy loam texture. The soil used to be a farmland and it has been abandoned since 2015. The soils were passed through a 2 mm sieve to remove stones and plants and mixed with sand (1:1, *v*:*v*). The mixed soils were then autoclaved at 121 °C for 2 h to remove live AMF spores. Cd chloride solution was added based on the dry weight of the soil to attain Cd concentrations of 0, 5, and 15 mg/kg in the soil. The soils were then mixed thoroughly and placed in equilibrium for one month to stabilize their chemical properties. Table 1 shows the separate parameters of the soil and sand prior to mixing and conducting the pot experiment.

### 2.2. Host Plants and AMF Inocula

Two plant species, *L. perenne* and *A. fruticosa*, were selected for use in this study. Their seeds were provided by the Muyang Gramineae Seed Technology Limited Company, China. The plant seeds were surface-sterilized in 75% alcohol (*v*:*v*) for 5 min before being rinsed five times with distilled water. They were then placed on wet filter paper in a Petri dish for germination. The AMF species *D. eburnea* used in the present study was supplied by the Bank of Glomeromycota in China, Beijing Academy of Agriculture and Forestry, China ([BGC] HK02C). The inoculum was obtained after six months of co-culture with corn (*Zea mays* L.) in autoclaved sand/vermiculite (1:1, *v*:*v*). It comprised a mixture of spores (40 spores g^−1^ inoculum), mycelia, root fragments, and soil.

### 2.3. Experimental Design

The experiment used a 2 × 3 × 2 factorial design with two plant species (*L. perenne* and *A. fruticosa*), three levels of Cd concentration (0, 5, and 15 mg/kg), and two AMF treatments (*D. eburnea* and a non-mycorrhizal control) in a completely randomized design. Each treatment had four replicates, yielding a total of 48 units (pots). Each plastic pot (upper diameter, 14 cm; lower diameter, 10 cm; height, 10 cm) was filled with 1 kg of soil. Before sowing the seeds, 30 g mycorrhizal inoculum (approximately 1200 spores) and soil were mixed and placed in a pot. For the non-mycorrhizal control treatment, 30 g of sterilized mycorrhizal inoculum was added.

A 10 mL AMF-free filtrate was applied to minimize differences in microbial communities between mycorrhizal and non-mycorrhizal pots. Thirty germinated seeds of *L. perenne* or ten germinated seeds of *A. fruticosa* were sown in each pot. After two weeks of germination, thinning was performed, leaving 15 seedlings for *L. perenne* and 5 seedlings for *A. fruticosa*. Seedlings were arranged under natural light and temperature conditions in a greenhouse and irrigated with tap water to maintain the soil at a water holding capacity of 70% during the growing period. The greenhouse temperature ranged from 18 °C to 34 °C during the day. For the first two weeks, pots received 10 mL of a full-strength Hoagland nutrient solution per week and were rotated weekly (Appendix A) [35]. The greenhouse pot experiments began on 23 August 2020, and ended on 22 October 2020.

### 2.4. Harvest and Measurements

At harvest, the plant shoots and roots were harvested separately. Roots were thoroughly cleaned with tap water to remove attached soil particles and then carefully cleaned with deionized water. Shoots and one root subsample were dried at 70 °C for 72 h and weighed. The AMF infection rate was determined using the other root subsample, and root biomass was calculated using the ratio of fresh and dry root weights. To determine the Cd concentration in the plants, the dried shoots and roots were ground and digested with HNO_3_–HClO_4_ (4:1, *v*:*v*). The digestion solution was fixed to 25 mL with 2% HNO_3_ and filtered after acid removal, and the Cd content was determined using graphite furnace atomic absorption spectrometry (GF-AAS, Perkin Elmer, Waltham, MA, USA). Soil pH was measured using a pH meter in a 1:2.5 (*w*:*v*) suspension of soil and deionized water. The NH_4_^+^-N and NO_3_^−^-N concentrations were measured using dual wavelength spectrophotometry and the indophenol blue method, and quantified using SmartChem 200 discrete auto analyzer (AMS Group Westco, Rome, Italy). Available phosphorus (A-P) was extracted using 0.5 M NaHCO_3_ at a pH of 8.5 and quantified using the SmartChem 200 discrete auto analyzer. The activities of soil alkaline phosphatase and neutral phosphatase were determined using respective assay kits (Solarbio, Beijing, China, BC0285, and BC0465). Soil Cd was separated into exchangeable Cd (1 M MgCl_2_, 2 h), carbonate bound Cd (1 M CH_3_COONa, 2 h), Fe-Mn oxide bound Cd (0.05 M NH_2_OH·HCl, 5 h), organic-bound Cd (0.02 M HNO_3_; 3.2 M NH_4_Ac, 2 h) and residual Cd (HF, HClO_4_ and HCl, high-purity, 0.5 h) fractions using Tessier’s five step extraction method [36]. Then, the contents of different Cd fractions were determined using GF-AAS.

The AMF structure of the roots was stained according to the improved Kormanik, method [37]. The cleaned fresh root segments (1 cm slices, 50 fragments randomly selected for each sample) were decolorized in 10% KOH solution (92 °C, 30 min). Roots were acidified with 2% HCl solution at 20 °C for 5 min and stained with 0.05% acid fuchsin solution (92 °C, 20 min). After the root fragments were stained, slides were prepared for observation. The structures of AM fungi in roots, such as arbuscule, vesicle, hyphal coil and non-septate hypha, were observed in each sample. The AMF colonization rate per unit root length was calculated using the grid-crossing method [38].

### 2.5. Calculation of Plant Cd Accumulation, Metal Tolerance Index, Transfer Factor, and Bioconcentration Factor

Plant Cd accumulation, metal tolerance index, transfer factor, and bioconcentration factor were used to evaluate the tolerance and Cd uptake capacity of plants. These indicators were calculated as follows:(1)Shoot Cd accumulation=shoot biomass × shoot Cd concentration
(2)Root Cd accumulation=root biomass × root Cd concentration
(3)$$\mathrm{Metal}\;\mathrm{tolerance}\;\mathrm{index}=\frac{\mathrm{plant}\;\mathrm{biomass}\;\mathrm{under}\;\mathrm{Cd}\;\mathrm{treatments}}{\mathrm{plant}\;\mathrm{biomass}\;\mathrm{under}\;\mathrm{Cd}\;\mathrm{free}\;\mathrm{soil}}$$
(4)$$\mathrm{Transfer}\;\mathrm{factor}=\frac{\mathrm{shoot}\;\mathrm{Cd}\;\mathrm{concentration}}{\mathrm{root}\;\mathrm{Cd}\;\mathrm{concentration}}$$
(5)$$\mathrm{Bioconcentration}\;\mathrm{factor}=\frac{\mathrm{total}\;\mathrm{Cd}\;\mathrm{concentration}\;\mathrm{in}\;\mathrm{plant}}{\mathrm{total}\;\mathrm{Cd}\;\mathrm{concentration}\;\mathrm{in}\;\mathrm{soil}}$$

### 2.6. Statistical Analysis

The two-way ANOVA was used to examine the effects of plant species, Cd, and their interaction on the *D. eburnea* colonization rate. The one-way ANOVA was used to examine the effects of AMF and Cd addition on soil properties, plant biomass, Cd uptake, and heavy metal tolerance for each plant species. Duncan’s test was used to compare the means among different treatments when the homogeneity of variance was satisfied; otherwise, the data were log or square root transformed. When the variances remained unequal, Tamhane’s T2 test was used as a post hoc test for multiple comparisons. A structural equation model (SEM) was also established using AMOS 24.0 to investigate the hypothetical approach associated with the effect of *D. eburnea* and Cd addition on the *D. eburnea* colonization rate and the bioconcentration factor for each plant species. Statistical analyses were performed using SPSS version 23.0, and the significance level was set at *p* < 0.05.

## 3. Results

### 3.1. D. eburnea Colonization Rate

The colonization rates of *D. eburnea* ranged from 11.00% to 29.60% for *L. perenne*, and 19.13% to 64.28% for *A. fruticosa* (Figure 1). The two-way ANOVA showed that the *D. eburnea* colonization rate was significantly influenced by the plant species and Cd, but not by their interaction (Figure 1). The addition of Cd had no effect on the root colonization rate of *L. perenne* (Figure 1). However, when compared with the Cd-free (0 mg/kg) treatment, the root colonization rate of *A. fruticosa* in low-Cd (5 mg/kg) and high-Cd (15 mg/kg) soil was significantly decreased by 45.15% and 26.10%, respectively (Figure 1).

### 3.2. Plant Biomass

Regardless of *D. eburnea* inoculation, the shoot biomass of *L. perenne* and *A. fruticosa* in the high-Cd treatment was lower than that in the Cd-free treatment (Figure 2a). Furthermore, in the absence of *D. eburnean*, *A. fruticosa* shoot biomass was significantly decreased in the low-Cd treatment. However, *D. eburnea* inoculation had no significant effect on shoot biomass of *L. perenne* or *A. fruticosa* in any of the Cd treatments (Figure 2a).

When compared with Cd-free plants, the root biomass of *L. perenne* was significantly increased by 62.81% in the high-Cd treatment when *D. eburnea* was present, while the total biomass of *L. perenne* was significantly decreased by 43.13% in the high-Cd treatment in the absence of *D. eburnea* (Figure 2b,c). *D. eburnea* inoculation significantly increased the root and total biomass of *L. perenne* by 206.47% and 103.72%, respectively, in the high-Cd treatment (Figure 2b,c). Regardless of *D. eburnea* inoculation, Cd addition had no significant effect on the root and total biomass of *A. fruticosa* (Figure 2b,c).

Without *D. eburnea* inoculation, the root-to-shoot rate of *L. perenne* was significantly higher under the low-Cd treatment than under the control and high-Cd treatments. By contrast, with *D. eburnea* inoculation, the root-to-shoot rate of *L. perenne* was significantly higher under the high-Cd treatment than under the Cd-free or low-Cd treatments (Figure 2d). However, Cd addition had no significant effect on the root to shoot rate of *A. fruticosa*, regardless of *D. eburnea* inoculation. *D. eburnea* inoculation significantly decreased the root to shoot rate of *L. perenne* by 27.31% under the low-Cd treatment, but significantly increased the root shoot ratio of *L. perenne* by 145.03% under the high-Cd treatment. Moreover, the root to shoot rate of *A. fruticosa* was significantly increased by *D. eburnea* inoculation by 58.21% and 77.00% under the control and low-Cd treatments, respectively (Figure 2d).

### 3.3. Soil Parameters

Compared with the Cd-free treatment, the pH of *L. perenne* soil was significantly increased in the low-Cd treatment without *D. eburnea* inoculation, whereas that of *A. fruticosa* soil was not significantly affected by *D. eburnea* inoculation or Cd addition (Table 2).

When *D. eburnea* was not inoculated, the available phosphorus (A-P) concentration in *L. perenne* soil in the low-Cd treatment was significantly lower than that under Cd-free or high-Cd treatments. However, compared with the Cd-free treatment, the A-P concentration in *L. perenne* soil was significantly increased by 96.70% under the low-Cd treatment when *D. eburnea* was inoculated. The A-P concentration in *A. fruticosa* soil was significantly increased in the high-Cd treatment, with or without *D. eburnea* (Table 2).

The high-Cd treatment significantly increased the NH_4_^+^-N and NO_3_^−^-N concentrations in *L. perenne* soil, compared to the low-Cd treatment after *D. eburnea* inoculation. However, Cd addition had no significant effect on the NH_4_^+^-N or NO_3_^−^-N concentrations in *A. fruticosa* soil, regardless of *D. eburnea* inoculation. *D. eburnea* inoculation had no significant effect on NH_4_^+^-N concentration in *L. perenne* or *A. fruticosa* soils in any of the Cd treatments (Table 2). However, *D. eburnea* inoculation significantly increased the NO_3_^−^-N concentration in *L. perenne* and *A. fruticosa* soils by 66.51% and 65.80%, respectively, under the high-Cd treatment (Table 2).

Neutral phosphatase activity in *L. perenne* and *A. fruticosa* soils and alkaline phosphatase activity in *L. perenne* soil were not significantly affected by Cd addition or *D. eburnea* inoculation (Table 2). Compared with Cd-free soil, the alkaline phosphatase activity in *A. fruticosa* soil was significantly increased by high-Cd treatment by 21.79% in the absence of *D. eburnea*, while it was significantly decreased in the high-Cd treatment by 24.35% in the presence of *D. eburnea* (Table 2). Alkaline phosphatase activity in *A. fruticosa* soil was significantly increased by *D. eburnea* inoculation under the Cd-free treatment, but was decreased by *D. eburnea* inoculation under the high-Cd treatment (Table 2).

### 3.4. Soil Cd Fractions

Compared with the low-Cd treatment, exchangeable Cd was significantly increased and carbonate-bound Cd was significantly decreased by the high-Cd addition in soils of both plant species, whether *D. eburnea* was inoculated or not (Figure 3a,b). Residual Cd was significantly higher under the low-Cd treatment without *D. eburnea* than under the other treatments (Figure 3a,b). Under the low-Cd treatment, *D. eburnea* inoculation significantly increased exchangeable Cd by 3.73% in *L. perenne* soil and significantly decreased residual Cd by 1.56% and 3.90% in *L. perenne* and *A. fruticosa* soils, respectively (Figure 3a,b).

### 3.5. Plant Cd Concentration and Accumulation

The shoot Cd concentrations of both plants significantly increased with an increase in the soil Cd concentration, regardless of *D. eburnea* inoculation (Figure 4a). *D. eburnea* inoculation significantly decreased the shoot Cd concentration of *L. perenne* by 28.54% under the high-Cd treatment, but had no significant effect on the shoot Cd concentration of *A. fruticosa* in any of the Cd treatments (Figure 4a). Regardless of *D. eburnea* inoculation, the high-Cd treatment significantly increased the root Cd concentration of *L. perenne* compared with the low-Cd treatment (Figure 4b). Among all the Cd treatments, *D. eburnea* inoculation had no significant effect on the root Cd concentration of *L. perenne*. Compared with the low-Cd treatment, the high-Cd treatment significantly increased the root Cd concentration of *A. fruticosa* by 189.74% under the *D. eburnea* inoculation treatment (Figure 4b). *D. eburnea* inoculation significantly increased the root Cd concentration of *A. fruticosa* by 252.44% under the high-Cd treatment (Figure 4b).

Regardless of *D. eburnea* inoculation, shoot Cd accumulation in *L. perenne* was significantly increased by the high-Cd treatment compared to the low-Cd treatment (Figure 4c). Neither *D. eburnea* inoculation nor Cd addition had a significant effect on shoot Cd accumulation in *A. fruticosa* (Figure 4c). Compared with the low-Cd treatment, the high-Cd treatment significantly increased root Cd accumulation in *L. perenne* under *D. eburnea* inoculation (Figure 4d). *D. eburnea* inoculation significantly increased root Cd accumulation in *L. perenne* by 260.98% under the high-Cd treatment. In addition, *D. eburnea* inoculation significantly increased root Cd accumulation in *A. fruticosa* by 161.37% and 398.31% under all low- and high-Cd treatments, respectively (Figure 4d).

### 3.6. Metal Tolerance Index, Transfer Factor, and Bioconcentration Factor (BCF)

In the absence of *D. eburnea*, the metal tolerance index of *L. perenne* was significantly reduced in the high-Cd treatment by 50.75% compared with the low-Cd treatment, but the metal tolerance index of *L. perenne* was significantly increased by 19.94% in the presence of *D. eburnea*. However, *D. eburnea* inoculation significantly decreased the metal tolerance index of *L. perenne* by 16.36% under the low-Cd treatment, but it significantly increased the metal tolerance index of *L. perenne* by 103.72% under the high-Cd treatment (Figure 5a). Compared with the low-Cd treatment, the metal tolerance index of *A. fruticosa* inoculated with *D. eburnea* in the high-Cd treatment was significantly reduced by 34.61%, and *D. eburnea* inoculation significantly increased the metal tolerance index of *A. fruticosa* by 33.30% under the low-Cd treatment (Figure 5a).

Compared with the low-Cd treatment, the high-Cd treatment had no significant effect on the transfer factor of *L. perenne*, but it significantly increased the transfer factor of *A. fruticosa* by 132.60% under non-AMF *D. eburnea* inoculation (Figure 5b), whereas *D. eburnea* inoculation significantly decreased the transfer factor of *A. fruticosa* by 59.34% under the high-Cd treatment (Figure 5b).

Compared with the low-Cd treatment, the bioconcentration factor of *A. fruticosa* was significantly increased by 8.77% in the high-Cd treatment by 8.77% in the presence of *D. eburnea*. *D. eburnea* inoculation had no significant effect on the bioconcentration factor of *L. perenne* but significantly increased the bioconcentration factor of *A. fruticosa* by 47.11% and 51.88% under the low and high Cd treatments, respectively (c).

### 3.7. Pathways of AMF D. eburnea and Cd Treatment Effects on D. eburnea Colonization Rate and Bioconcentration Factor

A structural equation model (SEM) analysis was conducted to separately evaluate the influence of the addition of AMF *D. eburnea* and Cd on the AMF colonization rate and bioconcentration factor for each plant species. According to the model, AMF *D. eburnea* inoculation had a direct and significant positive impact on the root biomass and plant colonization rate (Figure 6a) and a direct and positive impact on the soil neutral phosphatase content, but the impact was not significant (Figure 6a). The addition of Cd indirectly reduced the colonization rate of *D. eburnea* by increasing the proportion of exchangeable Cd, but it had no direct or significant impact on the colonization rate of *D. eburnea* (Figure 6a). Furthermore, *D. eburnea* inoculation had a direct and significant positive impact on the root biomass of *L. perenne* (Figure 6b), where the addition of Cd indirectly improved the bioconcentration factor of *L. perenne* by increasing the proportion of exchangeable Cd in the soil (Figure 6b). AMF *D. eburnea* inoculation had a direct and significant positive effect on the bioconcentration factor of *A. fruticose*, where the addition of Cd had a direct and significant negative impact on the root biomass of *A. fruticosa* (Figure 6c). Moreover, the addition of Cd indirectly reduced the root biomass of *A. fruticosa* by increasing the soil pH value (Figure 6c).

## 4. Discussion

### 4.1. Cd in Soil Affected the Colonization Rate of AMF D. eburnea

The colonization rate of AMF *D. eburnea* was decreased with an increase in Cd concentration in soil, and this occurrence has already been verified in other studies [39,40]. The structural equation model proved that Cd increased the proportion of exchangeable Cd in the soil, which increased the biological toxicity of Cd in the soil and inhibited the development of *D. eburnea* [41]. Moreover, the AMF colonization rate of Leguminosae *A. fruticosa* was generally higher than that of Gramineae *L. perenne*. It is considered that AMF provides *A. fruticosa* with plenty of phosphorus in the process of nitrogen fixation [33,34], while receiving nitrogen nutrition to improve symbiosis with the host plants [42].

### 4.2. AMF Inoculation and Cd Addition Affected Cd Fractions in Soil

The form of Cd in the soil affects its toxicity and uptake by plants [43]. In this study, the addition of Cd significantly increased the proportion of exchangeable Cd and reduced the proportion of carbonate bound Cd. Previous studies have shown that an increase in pH reduces the number of Cd ions in soil [44]. However, when pH increases to a high level, the solubility of Cd is increased [45,46,47], and some of the Cd ions in high-pH soil form Cd hydroxide precipitation [48,49]. In the present study, the AMF treatment increased the proportion of exchangeable Cd and decreased the residual Cd in the soil. This phenomenon has also been observed in previous studies [20]. It is considered that this occurred because *D. eburnea* inoculation increased the plant root biomass, and the amount of plant root activated soil Cd during the plant growth process, which resulted in an increase in the proportion of exchangeable Cd [50].

### 4.3. AMF Impact on Plant Biomass and Cd Uptake

*D. eburnea* inoculation increased the total biomass and root shoot ratio of both *L. perenne* and *A. fruticosa* under the high-Cd treatment, and this may have occurred because *D. eburnea* improved the soil environment (e.g., NO_3_^−^-N) and promotes plant growth [15]. The results of the structural equation model showed that *D. eburnea* inoculation had a positive effect on the root biomass, and the increased soil neutral phosphatase content significantly increased the plant root biomass (Figure 6a). The A-P concentration in soil was decreased in this study when *D. eburnea* was inoculated with a high-Cd treatment. This may be that the content of available phosphorus in the soil itself was relatively low (Table 1 and Table 2), and because *D. eburnea* inoculation increased the root biomass and the soil neutral phosphatase content, which enabled greater amounts of A-P to be absorbed by the plants [51]. Under the high-Cd treatment, *D. eburnea* reduced the transfer factor by 50.08% for *L. perenne* and 59.34% for *A. fruticosa*, which showed that AMF inhibited the transport of Cd to the plant aerial parts at high-Cd concentrations, thus reducing the damage caused by Cd to aerial parts and improving the resistance of plants. This phenomenon has also been observed in other studies [52,53], and it could result from AMF changing the intracellular distribution and chemical form of Cd in the roots and it increases the cellulose in root cell walls to bind Cd ions [25,54].

Some studies have shown that AMF can promote the absorption of Cd by plants [20,55], but other studies have reported opposite results [12,56]. The mechanism underlying this difference remains unclear. However, according to the findings of our study and those of others, it is possible that different AMF species, plant species, soil properties, and Cd concentrations have different effects on the efficiency of AMF in promoting Cd uptake. The differences between the plant species in this study were mainly evident with respect to the metal tolerance of *L. perenne* and *A. fruticosa*. In this respect, *L. perenne* showed a high tolerance to heavy metals in low-Cd soils and a low tolerance in high-Cd soils, whereas *A. fruticosa* showed a low tolerance to heavy metals in all of the Cd-treated soils, and the results are consistent with those of previous studies [57]. In summary, inoculation with *D. eburnea* increased the biomass of plant roots, increased the Cd content of plant roots, and improved the enrichment capacity of plants.

## 5. Conclusions

In our study, the biomass and Cd absorption capacity of *L. perenne* were higher than those of *A. fruticosa* regardless of *D. eburnea* inoculation. Under the high-Cd treatment, *D. eburnea* increased the root biomass and root shoot ratio of both *L. perenne* and *A. fruticosa*, and it also increased the Cd absorption and bioconcentration factor of *L. perenne* and *A. fruticosa* by altering the soil Cd speciation by increasing exchangeable Cd and decreasing residual Cd. However, Cd was mainly accumulated in the plant roots. Our study indicates that the combined use of *D. eburnea* and remediation plants may be a viable alternative for remediating Cd-contaminated soil. However, the current data were obtained under greenhouse conditions over a two-month period, and this may not be representative of the influence of AMF in a natural environment. In addition, soil heavy metal pollution under natural conditions can involve significant heavy metal content. Therefore, it is necessary to verify whether the results obtained in this study can be extrapolated to other heavy metal pollutants.

## Figures and Tables

**Figure 1 ijerph-20-00795-f001:**
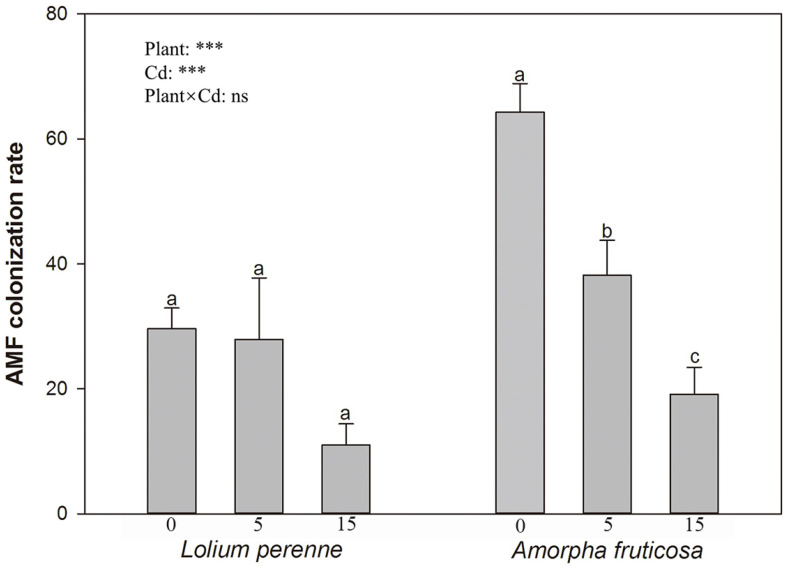
AMF colonization rate (Means ± SE, *n* = 4) of *L. perenne* and *A. fruticosa* under different Cd concentrations. Different letters indicate significant differences among treatments for each plant species according to Duncan’s test at *p* < 0.05. 0, no Cd added to soil; 5, 5 mg/kg Cd added to soil; 15, 15 mg/kg Cd added to soil. ***, *p* < 0.001; ns, not significant.

**Figure 2 ijerph-20-00795-f002:**
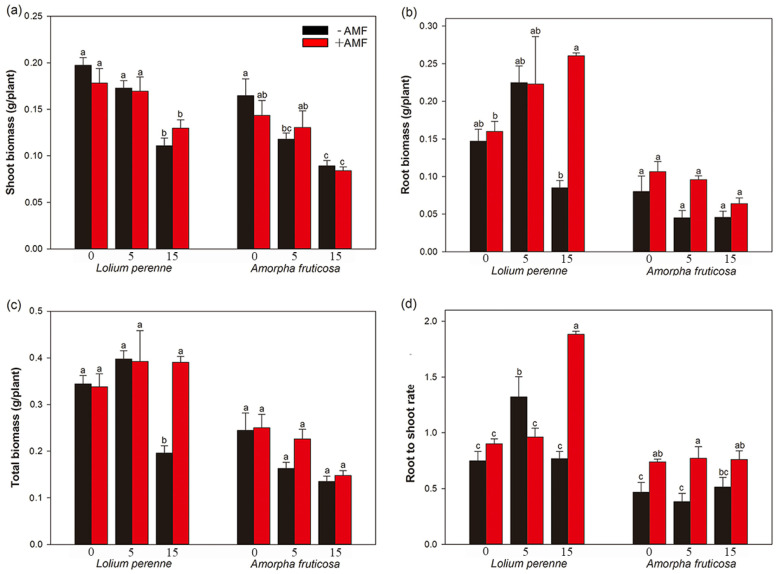
Effects of AMF and Cd addition on shoot biomass (**a**), root biomass (**b**), total biomass (**c**), and root to shoot rate (**d**) of *L. perenne* and *A. fruticosa* (Means ± SE, *n* = 4). Different letters indicate significant differences among treatments for each plant species according to Duncan’s or Tamhane’s T2 test at *p* < 0.05. −AMF, not inoculated with *D. eburnea*; +AMF, inoculated with *D. eburnea*; 0, no Cd added to soil; 5, 5 mg/kg Cd added to soil; 15, 15 mg/kg Cd added to soil.

**Figure 3 ijerph-20-00795-f003:**
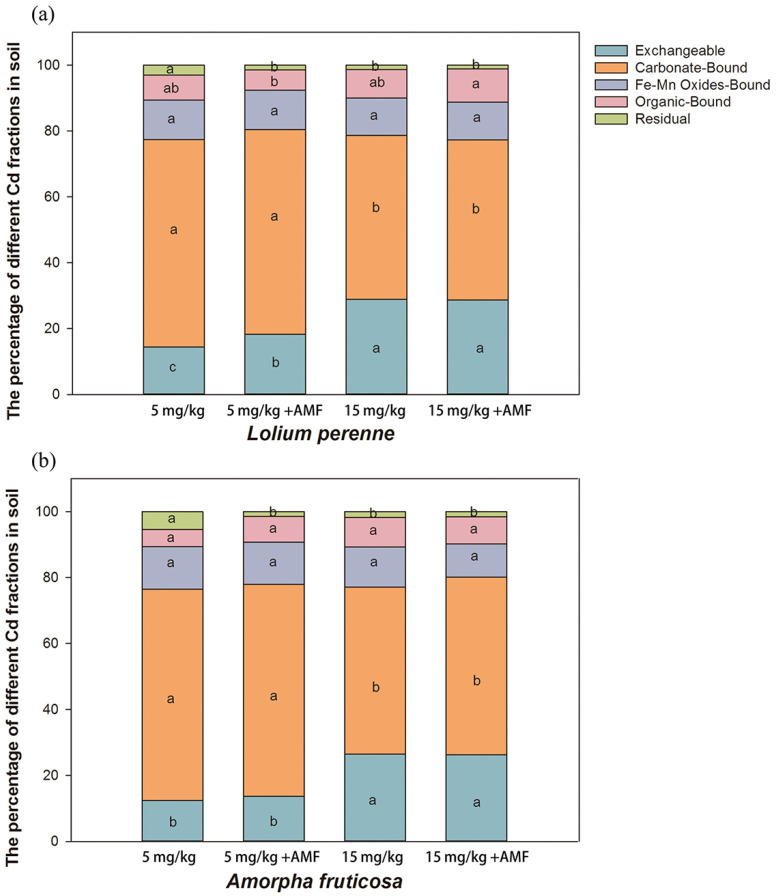
Percentage of different soil Cd forms under different treatments for *L. perenne* (**a**) and *A. fruticosa* (**b**) (means ± SE, *n* = 4). Different letters indicate significant differences among treatments for each plant species according to Duncan’s test or Tamhane’s T2 post hoc test at *p* < 0.05. 5 mg/kg, 5 mg/kg Cd added to soil; 5 mg/kg + AMF, 5 mg/kg cadmium, and *D. eburnea* added to soil; 15 mg/kg, 15 mg/kg Cd added to soil; 15 mg/kg + AMF, 15 mg/kg cadmium, and *D. eburnea* added to soil.

**Figure 4 ijerph-20-00795-f004:**
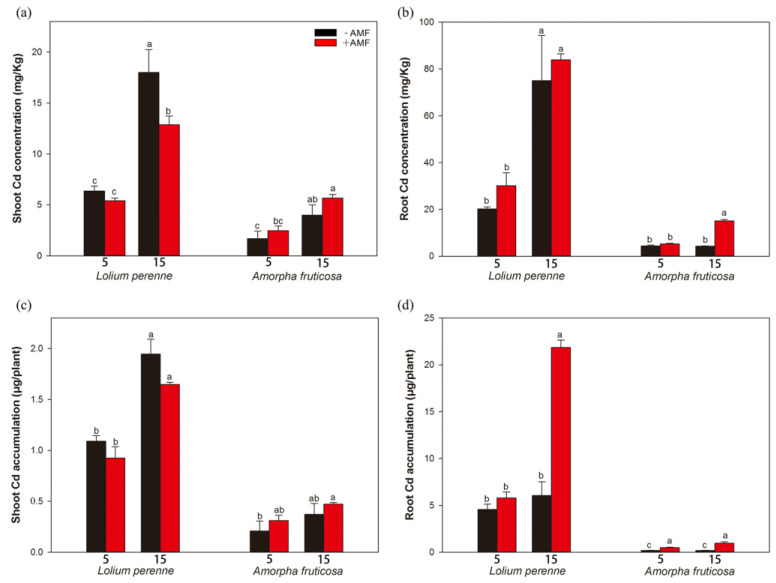
Effects of *D. eburnea* and Cd addition on shoot Cd concentration (**a**), root Cd concentration (**b**), shoot Cd accumulation (**c**), and root Cd accumulation (**d**) of *L. perenne* and *A. fruticosa* (Means ± SE, *n* = 4). Different letters indicate significant differences among treatments for each plant species according to Duncan’s test or Tamhane’s T2 post hoc test at *p* < 0.05. −AMF, not inoculated with *D. eburnea*; +AMF, inoculated with *D. eburnea*; 5, 5 mg/kg Cd added to soil; 15, 15 mg/kg Cd added to soil.

**Figure 5 ijerph-20-00795-f005:**
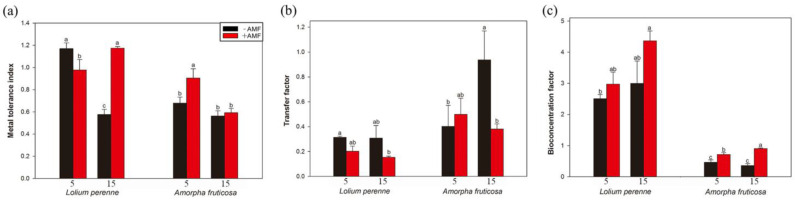
Effects of *D. eburnea* and Cd concentration on the metal tolerance index (**a**), transfer factor (**b**), and bioconcentration factor (**c**) of *L. perenne* and *A. fruticosa* (Means ± SE, *n* = 4). Different letters indicate significant differences among treatments for each plant species according to Duncan’s test or Tamhane’s T2 post hoc test at *p* < 0.05. −AMF, not inoculated* D. eburnea*; +AMF, inoculated* D. eburnea*; 5, 5 mg/kg Cd added to soil; 15, 15 mg/kg Cd added to soil.

**Figure 6 ijerph-20-00795-f006:**
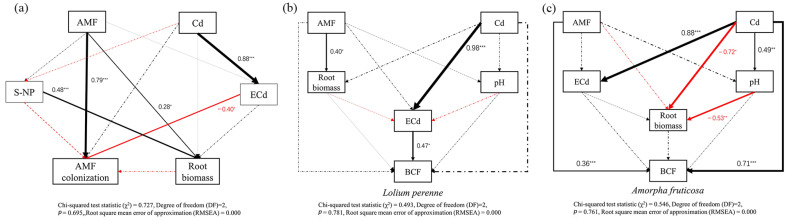
Structural equation model of the effects of AMF *D. eburnea* and Cd addition on the *D. eburnea* colonization rate (**a**), BCF of *L. perenne* (**b**) and *A. fruticosa* (**c**). The solid line indicates a significant correlation, while the dotted line indicates an insignificant correlation. The black and red arrows represent positive and negative effects, respectively. Values associated with the arrows and the arrow width represent standardized path coefficients. S-NP, soil neutral phosphatase, total biomass; ECd, exchangeable Cd; BCF, bioconcentration factor. *, *p* < 0.05; **, *p* < 0.01; ***, *p* < 0.001.

**Table 1 ijerph-20-00795-t001:** Initial parameters of the soil and sand used for the pot experiment (means ± SE, *n* = 3). MBC, microbial biomass carbon; MBN, microbial biomass nitrogen; A-P, available phosphorus.

Parameters	Soil Properties	Sand Properties
pH	8.43 ± 0.01	8.30 ± 0.01
MBC (mg/kg)	18.39 ± 0.09	19.26 ± 0.87
MBN (mg/kg)	3.61 ± 0.08	2.01 ± 0.07
A-P (mg/kg)	0.89 ± 0.04	1.51 ± 0.11
NH_4_^+^-N (mg/kg)	19.69 ± 0.57	12.15 ± 0.34
NO_3_^−^-N (mg/kg)	5.94 ± 0.14	3.67 ± 0.08
SOC (g/kg)	13.6 ± 0.05	4.01 ± 0.06
TC (g/kg)	15.10 ± 0.41	13.43 ± 0.24
TN (g/kg)	1.10 ± 0.20	3.98 ± 0.01
Total Cd (mg/kg)	0.87 ± 0.01	0.35 ± 0.01

**Table 2 ijerph-20-00795-t002:** Effects of arbuscular mycorrhizal fungi and Cd on soil properties.

	Treatment	CK	AMF	Cd5	Cd5 + AMF	Cd15	Cd15 + AMF
*L. perenne*	pH	9.44 ± 0.02 b	9.54 ± 0.05 ab	9.59 ± 0.02 a	9.54 ± 0.07 ab	9.51 ± 0.02 ab	9.49 ± 0.02 ab
	Available phosphorus (mg/kg)	8.07 ± 0.9 a	2.88 ± 0.5 c	1.47 ± 0.18 c	5.66 ± 1.4 ab	5.64 ± 0.9 ab	3.75 ± 0.77 bc
	NH_4_^+^-N (mg/kg)	25.45 ± 2.62 ab	23.82 ± 1.04 b	24.45 ± 2.22 ab	27.22 ± 1.63 ab	25.95 ± 2.11 ab	30.21 ± 1.46 a
	NO_3_^−^-N (mg/kg)	0.44 ± 0.14 b	0.79 ± 0.05 b	0.72 ± 0.12 b	0.99 ± 0.24 b	0.91 ± 0.12 b	1.52 ± 0.26 a
	Neutral phosphatase (U/g)	0.1 ± 0.01 a	0.11 ± 0.01 a	0.1 ± 0.0042 a	0.08 ± 0.01 a	0.08 ± 0.01 a	0.08 ± 0.01 a
	Alkaline phosphatase (U/g)	0.15 ± 0.01 a	0.16 ± 0.01 a	0.14 ± 0.01 a	0.15 ± 0.01 a	0.15 ± 0.01 a	0.15 ± 0.01 a
*A. fruticosa*	pH	9.23 ± 0.08 a	9.3 ± 0.04 a	9.42 ± 0.02 a	9.43 ± 0.04 a	9.37 ± 0.02 a	9.48 ± 0.04 a
	Available phosphorus (mg/kg)	1.5 ± 0.15 b	2.11 ± 0.18 b	1.98 ± 0.57 b	2.16 ± 0.27 b	6.46 ± 0.51 a	5.96 ± 0.67 a
	NH_4_^+^-N (mg/kg)	29.35 ± 2.85 a	30.08 ± 0.65 a	31.88 ± 1.32 a	31.7 ± 2.09 a	29.82 ± 1.06 a	37.53 ± 4.58 a
	NO_3_^−^-N (mg/kg)	4.3 ± 0.52 ab	6.51 ± 0.61 a	1.74 ± 0.47 b	5.13 ± 0.8 ab	1.73 ± 0.17 b	2.87 ± 0.52 ab
	Neutral phosphatase (U/g)	0.06 ± 0.001 a	0.06 ± 0.0029 a	0.07 ± 0.01 a	0.07 ± 0.01 a	0.07 ± 0.01 a	0.06 ± 0.0018 a
	Alkaline phosphatase (U/g)	0.14 ± 0.01 b	0.18 ± 0.01 a	0.16 ± 0.01 ab	0.15 ± 0.01 ab	0.17 ± 0.01 a	0.14 ± 0.01 b

Data represent the mean of four replicates ± SE. Different letters in each row indicate significant differences among treatments for each plant species according to Duncan’s test or Tamhane’s T2 post-hoc test at *p* < 0.05. CK, control; AMF, *D. eburnea* added to soil; Cd5, 5 mg/kg cadmium added to soil; Cd5 + AMF, 5 mg/kg cadmium, and *D. eburnea* added to soil; Cd15, 15 mg/kg cadmium added to soil; Cd15 + AMF, 15 mg/kg cadmium and *D. eburnea *added to soil.

## Data Availability

Applicable.

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
