# Peer review of "Effect of Arbuscular Mycorrhiza Fungus Diversispora eburnea Inoculation on Lolium perenne and Amorpha fruticosa Growth, Cadmium Uptake, and Soil Cadmium Speciation in Cadmium-Contaminated Soil"

_ijerph, 2023, doi:10.3390/ijerph20010795_

Round 1
Reviewer 1 Report
Ijerph-2048983
Effect of arbuscular mycorrhizal inoculation on Lolium perenne and Amorpha fruticose growth, cadmium uptake, and soil cadmium speciation in cadmium-contamined soil
Sun et al.
The authors investigated the effects of AMF on plant growth, Cd uptake, and Cd speciation in a contamined soil by using two AMF inoculants, three levels of Cd, and two plant species in a pot experiment. They have provided an interesting dataset and some evidence that AMF inoculation of plants in an alternative pathway for the remediation of Cd-contamined soils. In general, the study fits well to the scope of the Environmental Research and Public Health and such data is valuable to the region, even to the world (if they will be able to make the data public in the end; line 424-425, "Data availability statement: Not applicable."). Their sampling effort was not impressive, with in total 48 units (pots) collected across 3 months (from August 2020 to October 2020). I would be happy to see this manuscript published. However, I have some concerns about the data analyses and the way of presentation of the study. I address the concerns below:
1. I am not convinced about the phrase "the effects of AMF on Cd speciation to influence Cd uptake" used in the paper. I think it is misleading. The "AMF" in fact was just one AMF species (Diversispora eburnea), thus the term “AMF” has to be used carefully. I would not recommend applying the term "AMF" here. The authors may rephrase AMF as Diversispora eburnea.
2. Why did the authors choose Diversipora eburnea spores as inoculum? They need to provide an statement about it. Is there scientific evidence about Cd remediation by D. eburnean? Is D. eburnea efficient to colonize both L. perenne and A. fruticola? D. eburnea are abundant in contamined soil by Cd?
3. Why dit the authors choose L. perenne and A. fruticola as host-plants? Authors need to give an explanation supported by scientific evidence about their preferences.
4. Since the sampling effort was not impressive, I would not recommend describe Cd speciation. I was not convinced about it. The authors did not describe soil texture. For example, clay soils have different biogeochemical pathways from sandy soils. So, soil texture in an important factor influencing Cd sorption, AMF activity, and rootability (growth and activity). What was the soil texture?
5. Time is an important factor regarding Cd speciation. Did 3 months a enough time to suppose AMF influence on Cd speciation? Last but not least, root traits (growth, activity, density of fine roots, and plant history) determines Cd uptake and root colonization by AMF.
6. Previous soil management practices are important to consider the geochemical pathways of element. I cannot find the soil history (previous crops, tillage, fertilization, limestone, herbicides, fungicides, insecticides, etc). Why did the author choose the soil from the Campus of Henan University, China? Is it free of Cd contamination?
7. Describe in detail the composition of the used “full-strength Hoagland nutrient solution”. An electronic supplementary material would help.
8. Authors did not describe the methods to determine the different forms of soil Cd. They need to provide this information.
9. The statistical analysis. I recommend performing SEM analyses with clear competitive hypotheses proposed (to compare data supports between models). In my opinion, soil parameters, plant traits, and Cd data are highly correlated with (and will contribute to) the "Cd remediation". Following the SEM, the authors should report some predictive models to estimate the three "response" parameters: soil, plant traits and Cd dynamics.
10. English, grammar, and misspelling. I would not like to pinpoint each typo, grammatical error or capitalised or lowercase letter in the manuscript, because I do not think this is the responsibility of the reviewer. However, I would highly recommend the authors to have a throughout revision of their writing and try to avoid such mistakes.
Following are specific comments/questions to the lines (L) indicated in the manuscript:
L39-47: The study investigated the effects of D. eburnea not AMF species. I would recommend to rephrase it by describing D. eburnea instead “AMF”, the importance of considering D. eburnea, the influence of D. eburnea on Cd instead the influence of AMF on soil P.
L48-59: Again, the authors describe AMF in general. They need to be specific. Their results showed the influence of D. eburnea, not AMF in general. What is the D. eburnea role in coping with Cd stress? Does D. eburnea change the form of soil Cd? Does D. eburnea helps plants to remove soil Cd?
L60: The purpose was to investigate the effect of D. eburnea, not AMF in general. Please adjust all the text. It is annoying all the time reading “AMF”. This study used D. eburnea spores! Thus, authors need to be specific as well.
L68-75: Add soil texture, and soil history.
L69 and Table 1: In fact, they did not describe physical properties. They have biochemical properties in the Table 1. What do MBC, MBN, A-P mean? Are they microbial biomass C, microbial biomass nitrogen, and available P, respectively? Describe them clearly in the text. Please, avoid the use of acronyms/abbreviations.
L79-87: When the plants (L. perenne and A. fruticosa) received the inoculum with D. eburnea?
L121-124: Here there is a core information about the Cd fractions. Rephrase it carefully. Describe each fraction separately.
L138-148: Add a structural equation model and predictive models as described before.
L151-152: Why did the authors describe it? The lack of root colonization in the non-inoculated treatment is not a result. It is an obligation, and it just describes the qa/qc of samples.
L160-164: Please use the Cd levels instead Cd0, Cd5, and Cd15. Adjust it in the whole text.
Table 2: There is no need to show the ANOVA outputs. Transfers it to an electronic supplementary material.
Results sections: I would recommend the use “D. eburnea” instead “AMF”
L207-216: Unfortunately, they did not show physical properties. Thus, I recommend adjusting it accordingly their findings.
Figures: In general, they need to have their resolution improved. Some items are hard to read inside figures.
L350-357: Cd decreased AMF activity or Cd decreased root biomass? With less roots, there is a linear reduction of root colonization. Thus, Cd presents a direct influence on root traits, and indirect effects on AMF root colonization. This kind of assumption could be fixed by using a SEM approach.
L358-372: The main hypothesis of this manuscript was supported by these findings? L369-372: Did the authors analysed root exudation? If not, it is a poor conjecture. They have data about soil pH. Thus, they can support their findings about soil acidification without using conjectures about root exudates.
L373-398: Again, the SEM approach would help the authors to build the relationship among D. eburnea (not AMF), plant traits, and Cd uptake. It is clear to me the reader that authors had difficulties to build this paragraph. For example, the subsection 4.3 does not agree with the subsection 4.1. They need to rephrase all discussion section after running the SEM approach.
L424-425: Hopefully the data could be open accessed.
Reviewer 2 Report
Cadmium is one of the contaminant of soil in the world. It is still the problem that several cereals are causative food of cadmium intake. This manuscript is very good approach for the reduction of soil concentration of cadmium using microbiome. It seems very useful for the reduction of soil content of cadmium. This manuscript can propose how to regulate soil contaminations.
It is just wondering the three points.
1. Why is the difference between the plant species?
2. What is the main mechanism about AMF can absorb the cadmium?
3. This phenomenon is specific for cadmium or can be useful for other metals such as arsenic or mercury?
